# Heart rate and age modulate retinal pulsatile patterns

Ivana Labounková [1,2], René Labounek [2], Radim Kolář[1], Ralf P. Tornow [3], Charles F. Babbs[4], Collin M. McClelland[5], Benjamin R. Miller [6] & Igor Nestrašil [2,7 ✉]

Theoretical models of retinal hemodynamics showed the modulation of retinal pulsatile patterns (RPPs) by heart rate (HR), yet in-vivo validation and scientific merit of this biological process is lacking. Such evidence is critical for result interpretation, study design, and (patho-) physiological modeling of human biology spanning applications in various medical specialties. In retinal hemodynamic video-recordings, we characterize the morphology of RPPs and assess the impact of modulation by HR or other variables. Principal component analysis isolated two RPPs, i.e., spontaneous venous pulsation (SVP) and optic cup pulsation (OCP). Heart rate modulated SVP and OCP morphology ($p_{FDR} < 0.05$; age modulated SVP morphology ($p_{FDR} < 0.05$). In addition, age and HR demonstrated the effect on between-group differences. This knowledge greatly affects future study designs, analyses of between-group differences in RPPs, and biophysical models investigating relationships between RPPs, intracranial, intraocular pressures, and cardiovascular physiology.

[1] Department of Biomedical Engineering, Brno University of Technology, Brno, Czech Republic. [2] Division of Clinical Behavioral Neuroscience, Department of Pediatrics, Masonic Institute for the Developing Brain, University of Minnesota, Minneapolis, MN, USA. [3] Department of Ophthalmology, Friedrich-Alexander University of Erlangen-Nuremberg, Erlangen, Germany. [4] Weldon School of Biomedical Engineering, Purdue University, West Lafayette, IN, USA. [5] Department of Ophthalmology and Visual Neurosciences, University of Minnesota, Minneapolis, MN, USA. [6] Department of Neurology, University of Minnesota, Minneapolis, MN, USA. [7] Center for Magnetic Resonance Research, Department of Radiology, University of Minnesota, Minneapolis, MN, USA. ✉email: nestr007@umn.edu

In vivo dynamic video-ophthalmoscopy (VO) provides a potential opportunity for a non-invasive and easily accessible evaluation of retinal hemodynamics. VO is inexpensive and well suited to become a diagnostic and disease monitoring tool for the real-time imaging of local microvascular blood supply and detecting various pathologies. VO is applicable in many fields of medicine such as ophthalmology (e.g., diabetic retinopathy[1], glaucoma[2–4]), neurology (e.g., Alzheimer disease[5], multiple sclerosis[6], stroke[7]), or cardiology (e.g., coronary heart disease[8], arterial stiffness[9–11], hypertension[12], diabetes[12,13]).

Blood flow[14,15], blood volume[4,16] and structural venous diameter changes[17–19] are the most commonly evaluated hemodynamic parameters from dynamic retinal imaging. The hemodynamic parameters are usually extracted in a manual or semi-automated fashion from specific morphological segments of a retinal vessel tree[16–18,20–23]. However, the reproducibility of the parameters remains a challenge due to the uniqueness of each individual's retinal vessel tree[24] and the bias introduced by a subjective inter-rater and inter-participant selection of analyzed morphological segments[25]. To increase the reproducibility of local retinal hemodynamics, we have recently implemented a blind source separation that automatically divided the optic nerve head (ONH) into 2-3 functionally distinct areas that emerged as specific retinal pulsatile patterns (RPPs)[26]. The reproducible RPPs were spontaneous venous pulsation (SVP) and optic cup pulsation (OCP) with mutually phase-shifted hemodynamics. OCP was postulated to represent arterial blood filling preceding the SVP outflow[26].

Spontaneous venous pulsation is the most investigated hemodynamic phenomenon over the whole retinal background with the best detection capability and highest reproducibility in the area where the central retinal vein crosses the ONH. The etiology and visibility of the ONH SVP are likely related to the gradient between intracranial pressure (ICP) and intraocular pressure (IOP) waveforms[27–30]. Limited in vivo evidence of the SVP-ICP relationship exists[30–34]. The underlying biophysics is a subject of active investigation[35].

Mathematical models, which were proposed to describe IOP-ICP-SVP relationships[35–38], have not been verified by in vivo experiments yet. Levine and Bebie's SVP-ICP theory assumes an influence of heart rate (HR) on SVP amplitude, yet without providing the in vivo evidence[37,38]. Therefore, the influence of HR on retinal intra-vessel hemodynamics deserves further investigation. In addition to HR, demographic characteristics (e.g., age or sex) were linked to vessel stiffness[39,40], vessel cross-section area[41,42], pulse pressure[39,40] or cardiac cycle parameters (e.g., filling time, preload, stroke volume) and, consequently, modify intra-vessel hemodynamics including the blood flow/volume in retinal mini- and micro-vessels[42]. Exactly how the in vivo video-ophthalmoscopic data (sensitive to blood flow or volume fluctuations) are affected by HR, age, or sex remains to be determined. If the impact of any variable is proved, a revisiting of clinical video-ophthalmoscopic outcomes demonstrating correlations between VO measurements and retinal neural fiber layer (RNFL) thickness[4,19,43] or other outcomes in populations of healthy subjects and those with disease conditions such as glaucoma is warranted.

In this study, we investigated human in vivo video-ophthalmoscopic data and the effects of HR and age on the morphology of blood-volume-specific RPPs. We isolated two RPPs (i.e., SVP and OCP) from the video-ophthalmoscopic dataset, tested SVP-OCP phase shift, evaluated RPP reproducibility and morphology, and cross-correlated the morphology with HR, age, IOP, and RNFL thickness. Finally, we estimated the effects of HR and age with between-group comparisons in resulting morphological observations.

## Results

**Participant characteristics.** Thirty-four retinal video-recordings (RVRs) were acquired, and exclusively left-eye RVRs were used in the analysis. HR estimated from SVP and OCP of all participants was $66 \pm 13 \, \text{min}^{-1}$ (ranging $44$–$92 \, \text{min}^{-1}$). (Image analysis workflow estimating and segregating SVP and OCP is summarized in Fig. 1.) HR was significantly higher in patients with treated ocular hypertension (OHT) than in healthy participants and was significantly correlated with SVP and OCP morphologies (Table 1, Figs. 2, 3). Due to this finding, HR was treated as a confounding variable in further ANCOVA between-group tests. Other physiological data such as age, IOP, and RNFL demonstrated no significant between-group differences (Table 1). Refractive error, visual acuity, and perimetry were within physiological ranges without any significant pathology.

**Phase shifted retinal pulsatile patterns in principal component space.** In a total of 34 RVRs, we detected SVP in 33 RVRs (97%) and OCP in 31 RVRs (91%) when the first 12 principal components were visually inspected and evaluated in each RVR. Averaged SVP principal component index was $3 \pm 2$ (min 1; max 9), and averaged OCP principal component index was $3 \pm 2$ (min 1; max 11). Representative examples of SVP and OCP spatiotemporal patterns with detected and corrected control points are shown in Fig. 1a. An input-output workflow of the utilized principal component analysis (PCA) is schematically summarized in Fig. 1b.

The cross-correlation response function (Fig. 1a) demonstrated that OCP significantly preceded SVP about $-3.71 \pm 2.05$ (min $-8$; max 0) samples, i.e., $148 \pm 82 \, \text{ms}$, (one sample t-test $p = 3.95e^{-11}$). With regard to the participant's heart cycle period, the preceding interval corresponded to $-16 \pm 8\%$ of the heart cycle. The Pearson correlation coefficient (r) between not-aligned OCP and SVP time-courses (i.e., corresponding PCA eigenvectors[44]; Fig. 1b) was $0.31 \pm 0.39$ (min $-0.45$; max 0.92) and increased after the delay alignment to $0.84 \pm 0.08$ (min 0.66; max 0.96). The linear dependence between PCA eigenvector time-course and averaged image-intensity computed from the same region of interest (ROI) corresponding to the PCA-suprathreshold ROI was $0.55 \pm 0.33$ (min $-0.06$; max 0.97) for SVP and $0.51 \pm 0.29$ (min $-0.11$; max 0.94) for OCP. Lower correlation corresponded to recordings where the averaging approach failed due to high random noise power.

**Retinal pulsatile pattern morphology modulated by heart rate and age.** A sketch of the morphological measurement from the segregated RPP is shown in Fig. 1a. Morphologies of SVP and OCP were significantly modulated by HR (Figs. 2, 3). SVP amplitude Ampl ($r = -0.61$, $p = 0.0002$, $p_{BH} = 0.0003$, $P_{size} = 97.8\%$; $p_{BH}$ - Benjamini-Hochberg adjusted p-value, $P_{size}$ – power of the result regarding the sample size), total relative pulse stroke volume $V_T$ ($r = -0.87$, $p < 0.0001$, $p_{BH} = 0.0002$, $P_{size} = 100.0\%$), slope-up to baseline SlpU ($r = 0.46$, $p = 0.0067$, $p_{BH} = 0.0084$, $P_{size} = 79.7\%$) and time to peak $t_p$ ($r = -0.67$, $p < 0.0001$, $p_{BH} = 0.0002$, $P_{size} = 99.5\%$) were significantly associated with HR (Fig. 2b). SVP slope-down to peak SlpD was HR independent (Fig. 2). OCP $V_T$ ($r = -0.53$, $p = 0.0029$, $p_{BH} = 0.0073$, $P_{size} = 91.1\%$), SlpU ($r = 0.48$, $p = 0.0078$, $p_{BH} = 0.0130$, $P_{size} = 83.5\%$), $t_p$ ($r = -0.61$, $p = 0.0004$, $p_{BH} = 0.0020$, $P_{size} = 97.8\%$) and SlpD ($r = -0.45$, $p = 0.0154$, $p_{BH} = 0.0193$, $P_{size} = 77.6\%$) were significantly correlated with HR (Fig. 3b). OCP Ampl was HR independent (Fig. 3). HR-dependent intra-participant variability is shown in Fig. 4 for three representative participants out of the four participants who underwent RVRs twice (once at monocular and one at binocular VO) with an inter-recording interval about 2 years.

HR significantly correlated with age ($r = -0.45$, $p = 0.0090$, $P_{size} = 77.6\%$). Therefore, we investigated linear dependence

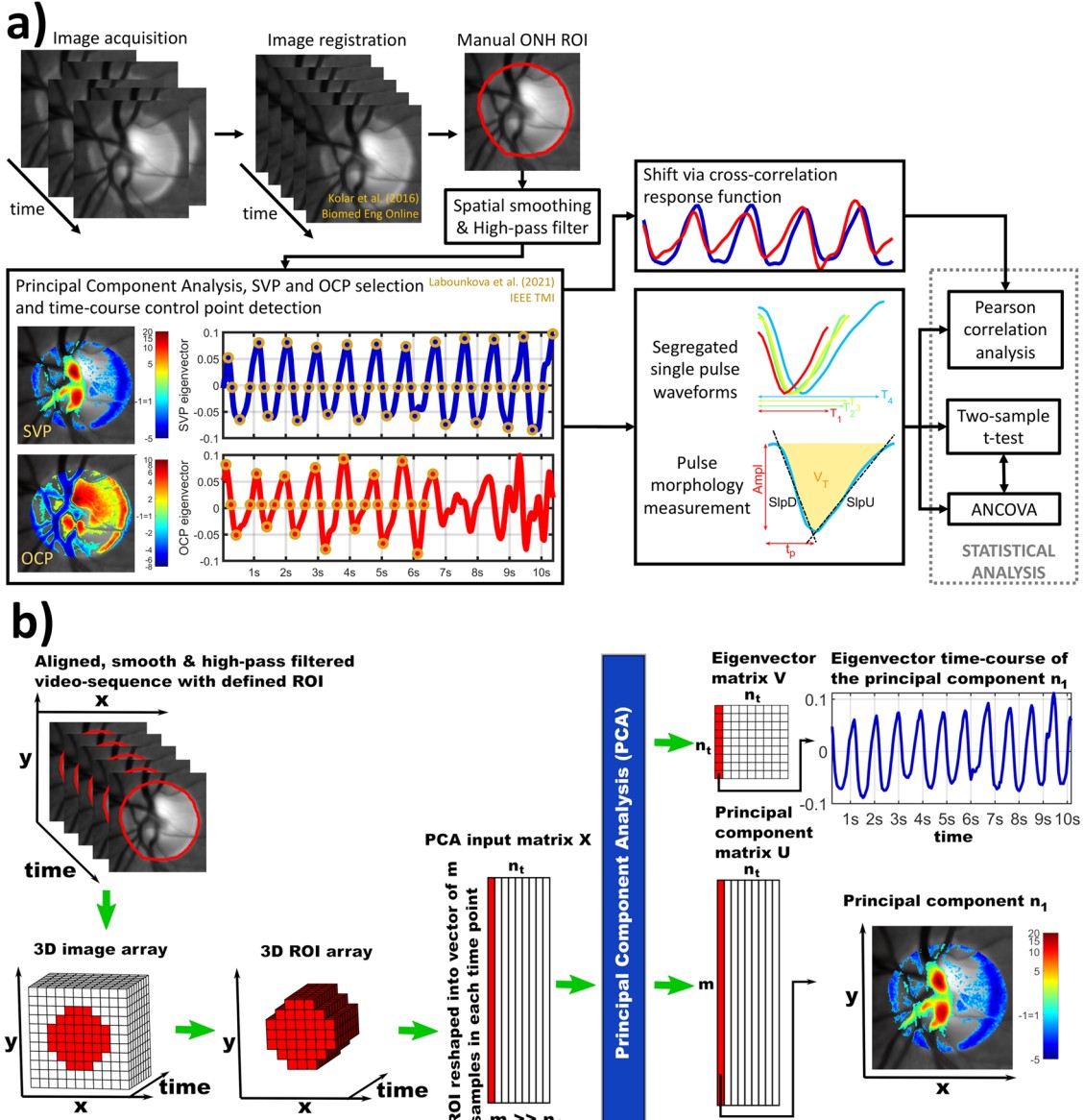

**Fig. 1 Image and statistical analysis workflow. a** The block diagram summarizes the whole image and statistical analysis workflow. ONH optic nerve head, ROI region of interest, SVP spontaneous venous pulsation, OCP optic cup pulsation, T period [s], Ampl amplitude, SlpD slope-down, SlpU slope-up, $t_p$ time to peak, $V_T$ total relative pulse stroke volume, ANCOVA analysis of covariance. **b** The block diagram summarizes the input-output system of the utilized principal component analysis (PCA). The left input side demonstrates array operations forming the PCA input matrix. The right output side demonstrates temporal and spatial extraction of the results. x, y spatial axes of images, $n_t$ total number of time points, m total number of pixels of the ROI in one time point, $n_1$ index of the principal component number 1 in the output principal component and eigenvector matrices (first red-highlighted columns), and time point number 1 in the PCA input matrix. Each column in the eigenvector matrix represents an eigenvector time-course of the principal component in the corresponding column of the principal component matrix. In the right bottom corner, the thresholded n1 principal component of m-samples was reshaped back into the original xy space and overlayed the averaged anatomical background image. All images show recordings of the left eye.

between RPP morphology and additional physiological variables. SVP Ampl ($r = 0.50$, $p = 0.0026$, $p_{BH} = 0.0130$, $P_{size} = 87.0\%$) and SVP $V_T$ ($r = 0.45$, $p = 0.0089$, $p_{BH} = 0.0223$, $P_{size} = 77.6\%$) were significantly correlated with age (Fig. 5). The modulation of SVP Ampl and $V_T$ demonstrated a larger effect of HR than age (Figs. 2, 5). No significant correlations of RPP morphology with IOP or RNFL were detected.

**Between-group differences in RPP morphology measurements.** Lower SVP Ampl, SVP $V_T$, and OCP $t_p$, along with higher absolute OCP SlpD and OCP SlpU, were detected in treated OHT patients (two-sample *t*-test) (Table 1). ANCOVA with age as a

confounding variable preserved all significant between-group differences in OCP morphology, but none in SVP morphology (Table 1). In contrast, ANCOVA with HR as a confounding variable only showed trends of the higher absolute OCP SlpD and the lower OCP $t_p$ (Table 1). Group averaged OCP pulses with 25–75% confidence intervals are shown in Fig. 6 and demonstrate visible changes in group-specific pulse morphologies.

## Discussion

**Study novelty and practical impact.** Our in vivo results demonstrate the modulation of the SVP and OCP pulse morphologies by HR and of the SVP morphology by age.

**Table 1 Participant characteristics and morphology measurements of retinal pulsatile patterns.**

| | | Age [y.o.] | HR [min⁻¹] | IOP [mmHg] | RNFL [μm] | |
|---|---|---|---|---|---|---|
| | | Age [y.o.] | HR [$min^{-1}$] | IOP [mmHg] | RNFL [μm] | |
| Physiology | p_ttest2 | 0.1177 | *0.0182 | 0.2198 | 0.5794 | |
| | Healthy controls | 66.0 ± 13.2 | 59.9 ± 9.6 | 15.3 ± 2.7 | 91.9 ± 10.5 | |
| | OHT patients | 58.7 ± 12.9 | 70.3 ± 13.5 | 16.5 ± 3.0 | 89.7 ± 11.6 | |
| | | Ampl | $V_T$ | SlpD | SlpU | $t_p$ [ms] |
| SVP | p_ttest2 | *0.0235 | *0.0317 | 0.4045 | 0.5110 | 0.3537 |
| | p_ANCOVA_HR | 0.2753 | 0.8049 | 0.2097 | 0.6353 | 0.3897 |
| | p_ANCOVA_age | 0.1057 | 0.1211 | 0.6646 | 0.5442 | 0.4962 |
| | Healthy controls | 0.143 ± 0.023 | 2.14 ± 0.62 | −0.587 ± 0.123 | 0.240 ± 0.061 | 419 ± 66 |
| | OHT patients | 0.123 ± 0.027 | 1.66 ± 0.62 | −0.544 ± 0.166 | 0.254 ± 0.059 | 396 ± 74 |
| OCP | p_ttest2 | 0.2105 | 0.8740 | *0.0077 | *0.0150 | *0.0047 |
| | p_ANCOVA_HR | 0.1311 | 0.1532 | 0.0637 | 0.1299 | 0.0859 |
| | p_ANCOVA_age | 0.1070 | 0.7891 | *0.0058 | *0.0199 | *0.0091 |
| | Healthy controls | 0.104 ± 0.035 | 1.34 ± 0.42 | −0.395 ± 0.199 | 0.174 ± 0.057 | 434 ± 81 |
| | OHT patients | 0.120 ± 0.031 | 1.31 ± 0.44 | −0.590 ± 0.167 | 0.247 ± 0.087 | 355 ± 56 |

*P*-values demonstrating between-group trends ($p < 0.1$) are highlighted with bold font, and *p*-values demonstrating significant between-group differences ($p < 0.05$) are highlighted with bold font and asterisks.
Notes: *OHT* ocular hypertension, *SVP* spontaneous venous pulsations, *OCP* optic cup pulsations, *HR* heart rate, *IOP* intraocular pressure, *RNFL* retinal neural fiber layer thickness, *Ampl* amplitude, $V_T$ total relative pulse stroke volume, *SlpD* slope-down, *SlpU* slope-up, $t_p$ time to peak, *p_ttest2* *p*-value of between-group difference obtained from the two-sample t-test, *p_ANCOVA_HR* *p*-value of between group-difference obtained from analysis of covariance where HR was confounding variable, *p_ANCOVA_age* *p*-value of between group-difference obtained from analysis of covariance where age was confounding variable. Values listed for HC and OHT are mean ± standard deviation. *P*-values demonstrating between-group trends ($p < 0.1$) are highlighted with bold font, and *p*-values demonstrating significant between-group differences ($p < 0.05$) are highlighted with bold font and asterisks. ANCOVA test, where two confounding variables (HR and age) would be used at once, has not been utilized because HR and age are significantly correlated pair of variables.

Therefore, the unverified biological impact of HR and age at RPP morphology was in vivo confirmed and validated. The impact of HR and age on the SVP/OCP morphology has been neglected in previous clinical research[4,19,33,34,43] and even in theoretical SVP-intracranial/intraocular pressure models[35,36]. The in vivo findings validate Levine and Bebie's theory, assuming that SVP Ampl is influenced by HR[37,38]. This discovery may also indicate descriptive vessel compliance characteristics as demonstrated by the uniformity of the SVP SlpD parameter. The studied morphology parameters should be investigated in future research as objective quantitative in vivo markers of retinal hemodynamics. Importantly, HR and age need to be considered and added as confounding variables in biological models or clinical studies evaluating between-group differences in RPP. The optimal strategy for the study design is a dataset free of between-group differences in HR and age, but such proper matching would be challenging and hardly achievable.

The analysis via spatial PCA of RVRs detected reproducible phase-shifted SVP and OCP patterns in healthy controls and treated OHT patients. SVP and OCP patterns demonstrated low inter-participant variation in eigenvector scaled time-courses, i.e., temporal pulses, and were highly reproducible. These results emphasize the feasibility of the spatial PCA for inferences achieved by RPPs and its applicability in future studies, especially those involving large population cohorts.

Although the etiology of the SVP phenomenon is still not clearly understood, the temporal uniformity in SVP SlpD can represent a novel insight into the SVP origin. This temporal uniformity may be due to similar inter-participant venous compliance characteristics that modulates venous capacitance. As the implication of vein capability to change its geometry, the vein compliance property enables a volume increase in local vein segments in response to local blood filling[45]. Since the vein resistance to local blood flow is minimal[45], the blood filling corresponding to SlpD in the SVP morphology displayed temporal uniformity over investigated population.

We revealed the significant impact of HR on the morphology of reproducible RPPs. The higher HR resulted in lower SVP Ampl, $V_T$, and $t_p$ and higher SlpU. Similarly, the higher HR led to the lower OCP $V_T$, $t_p$, and SlpD and higher SlpU. This observation of changes in $V_T$ linked to HR may reflect the

impact of Starling's law. Higher HR causes a shorter period for cardiac blood filling and, consequently, lower cardiac preload leads to the lower blood volume ejected from the heart. All these characteristics of heart function imprint into the RPP morphology and may be evaluated non-invasively and in vivo with the VO.

Additionally, three SVP parameters, i.e., Ampl, SlpD, and $V_T$, were significantly correlated with age. Higher SVP Ampl and $V_T$ with age can be influenced by peripheral pulse pressure pulsatility or vessel stiffening, which increase with age[11,46]. The increase of age-related vessel stiffness is directly linked to the elasticity loss[39,40] and is a potential factor underlying the steeper SVP SlpD trend observed in our older participants. Mechanistically, lesser venous capacitance leads to faster venous volume outflow. If the hypothesis is true, the SVP SlpD may become a non-invasive and fully automated measure proportional to vessel stiffness.

Previous studies showed that SVP were influenced by the IOP[28,35,37,47] and RNFL significantly correlated with RPP morphological measurements in the dataset, including population with retinal neurodegenerative disease[4,19,43]. In our participants, the IOP and RNFL did not correlate with RPP morphology. A potential explanation may lie in well-controlled OHT condition in our patient's cohort who presented with normalized IOP, so no neurodegeneration assessed by RNFL was present.

SVP morphology is believed to originate from a gradient between IOP and ICP, and many theoretical models attempted to express the SVP-IOP-ICP relationships[35–38]. However, the majority of previously reported models have not accounted for the effects of HR or age on the SVP Ampl or timings[35–38]. Therefore, the current in vivo observation represents critical information that adds another piece to the puzzle of the SVP etiology conundrum. Therefore, future models and studies evaluating SVP should account for the effects of HR and age to avoid potential erroneous conclusions.

Similar conclusions apply to the OCP assessment. Although OCP morphology demonstrated significant differences between healthy controls and OHT patients, a considerable deal of uncertainty remains as the OCP morphology changes were HR-related. The ANCOVA demonstrated the HR effect on between-group comparisons. It is important to note that the impact of HR has usually been omitted in previous clinical studies. In particular,

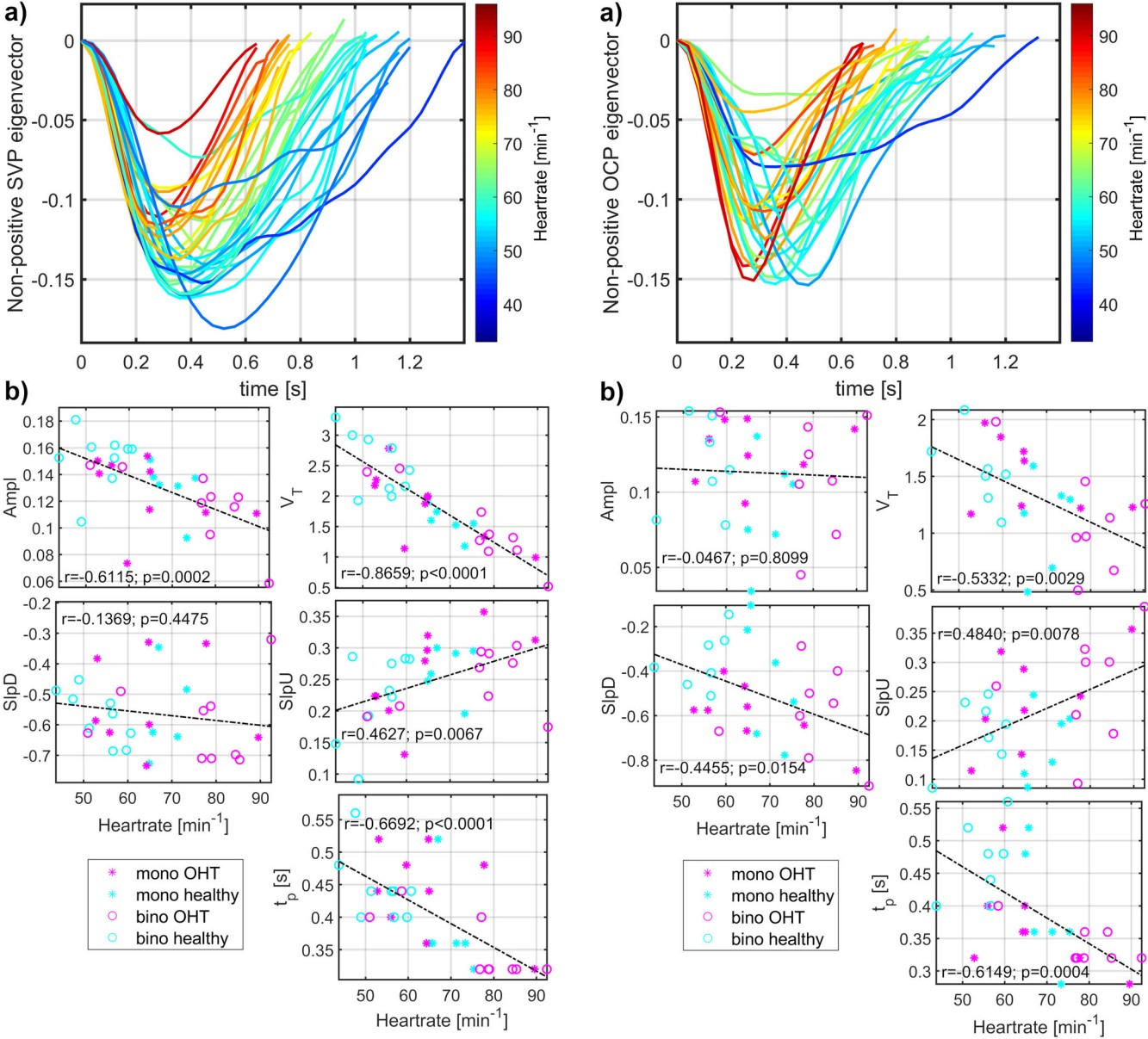

**Fig. 2 Heart rate modulated morphology of spontaneous venous pulsations (SVP).** **a** Visualization of mean single-pulses over all monocular (mono) and binocular (bino) retinal video-recordings from healthy controls and patients with medicated ocular hypertension (OHT). Graph lines are heart rate color-coded. **b** Evaluation of linear dependence between heart rate and SVP morphology measurements (Ampl amplitude of the SVP eigenvector, $V_T$ total relative pulse stroke volume in the eigenvector measures, SlpD slope-down from the eigenvector value at the period beginning to the negative eigenvector peak ≈ peak of the maximal absolute blood volume time-point, SlpU slope-up from the negative eigenvector peak to the period end, $t_p$ time to the negative eigenvector peak). Value r represents a corresponding Pearson correlation coefficient and value $p$ the $p$-value of the correlation level.

**Fig. 3 Heart rate modulated morphology of optic cup pulsations (OCP).** **a** Visualization of mean single-pulses over all monocular (mono) and binocular (bino) retinal video-recordings from healthy controls and patients with medicated ocular hypertension (OHT). Graph lines are heart rate color-coded. **b** Evaluation of linear dependence between heart rate and OCP morphology measurements (Ampl amplitude of the OCP eigenvector, $V_T$ total relative pulse stroke volume in the eigenvector measures, SlpD slope-down from the eigenvector value at the period beginning to the negative eigenvector peak ≈ peak of the maximal absolute blood volume time-point, SlpU slope-up from the negative eigenvector peak to the period end, $t_p$ time to the negative eigenvector peak). Value r represents a corresponding Pearson correlation coefficient and value $p$ the $p$-value of the correlation level.

ICP studies[33,34] (using Hedges scale[48]) or glaucoma diagnostic studies[4,19] demonstrated qualitative or quantitative alterations in RPP morphology without considering group-specific HR distributions. Future studies may avoid potentially blurred and inaccurate outcomes by reporting or, ideally, accounting for the between-group differences in HR.

Although the PCA is an automated method, the presented data processing pipeline utilized three manual interventions with the

potential impact on outcome measures. The interventions were: (i) manual ONH segmentation; (ii) identification of SVP and OCP components; and (iii) control point corrections. As PCA integrates eigen time-course for each principal component estimated from the whole ONH area, minimal imperfections in ONH segmentation should not crucially impact outcome measures. But, a further study investigating the PCA method robustness is warranted to clarify this matter. The automated

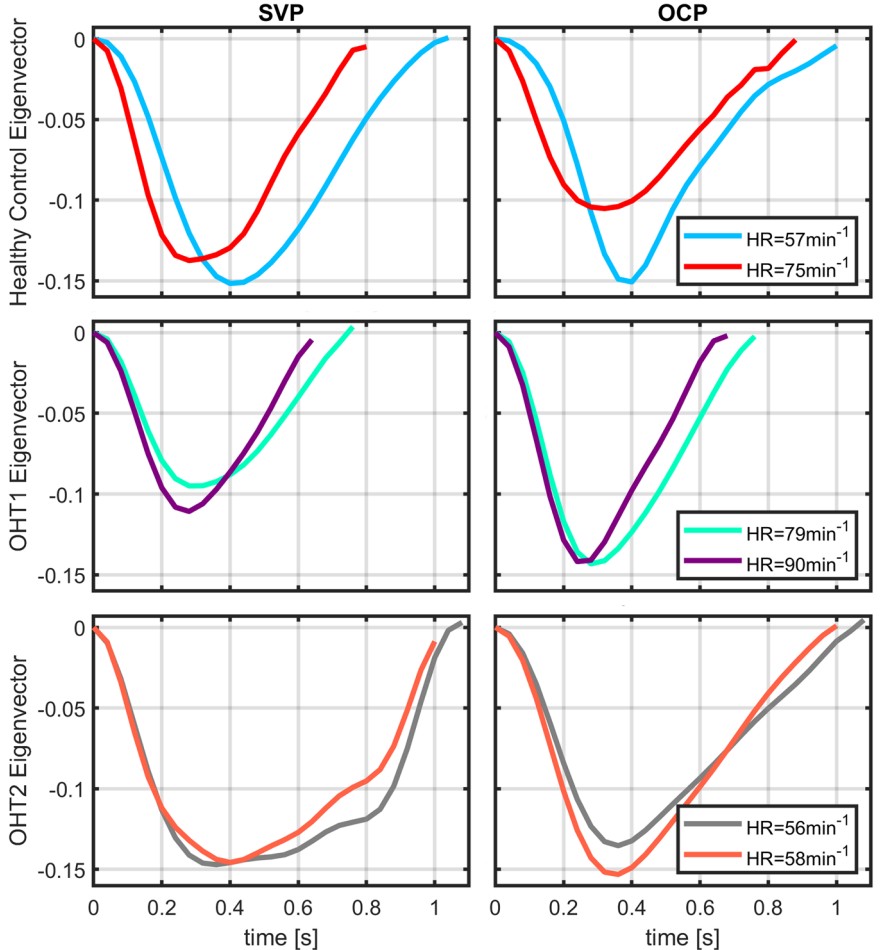

**Fig. 4 Representative examples of heart rate (HR) dependent intra-participant variability of retinal pulsatile patterns in one healthy control and two medicated ocular hypertension (OHT) patients.** For each participant, one retinal video-recording (RVR) was acquired with the monocular video-ophthalmoscope (VO) utilizing the CCD camera chip and one with the binocular VO utilizing the CMOS camera chip. The between-acquisition time interval was about two years for each representative participant. In healthy control and one OHT participant (OHT1), intra-participant RPP morphologies were dissimilar at different HR while RPP morphology remained unchanged for comparable HR as captured for other OHT participant (OHT2). Another OHT participant had different HR over acquisitions with similar outcomes as presented for the OHT1 participant. SVP spontaneous venous pulsations, OCP optic cup pulsations, HR heart rate.

identification of components of interest represents a challenge. We are working on the automatization of SVP and OCP identification. A wrong component selection can impact outcome measures so that morphology values are evaluated for a wrong pattern. Finally, isolated manual corrections of control points improve the precision of the averaged single pulse estimation and the precision of outcome measures.

Our study has several limitations to be addressed in future research. The sample size of our dataset is limited and should be extended to reproduce and validate our pilot observations. Still, the utilized correction for multiple comparison errors and the power analysis support that the presented correlation measures are related to the human body physiology rather than false positive observations. An optimal re-test should involve hundreds of participants with RVRs acquired at various video-ophthalmoscopes of various assembling technologies. Therefore, we would like to initialize a multi-center RVR challenge collecting 10 s RVRs to re-test the impact of HR and age at RPP morphology. Four participants who had both monocular and binocular RVR, approximately two years apart, may decrease inter-participant variability in our dataset. We consider this effect of being minimal as HR differed between the first and second RVR in three of four participants and as the intra-participant

variability mostly followed dataset trends of correlation measurements. Pharmacological treatment of OHT patients may have influenced the morphology measurements, as specific drugs can alter ocular hemodynamics. The high diversity of used drugs in our OHT group and our small sample size prevented the testing post-hoc differences between the untreated group and treated subgroups (e.g., beta-blocker versus prostaglandin treatment). The differences in vessel stiffness and pulse pressure related to sex[49,50] may also play a significant role in RPP morphology. Sex effects need to be validated in the larger cohort of participants.

In conclusion, we have demonstrated the in vivo evidence that heart rate and age modulate the morphology of retinal pulsatile patterns in humans. The observation corroborates Levine and Bebie's theory. The presented study will impact the design of future biological and clinical studies, future analyses of between-group differences in morphology of RPPs, and SVP-ICP-IOP biophysical modeling by emphasizing the necessity to include heart rate and age as important confounding factors.

## Methods
**Experimental design**. In concordance with the Declaration of Helsinki, all participants signed an informed consent approved by the ethical committee at the Friedrich-Alexander University of Erlangen-Nürnberg. The participants are part of

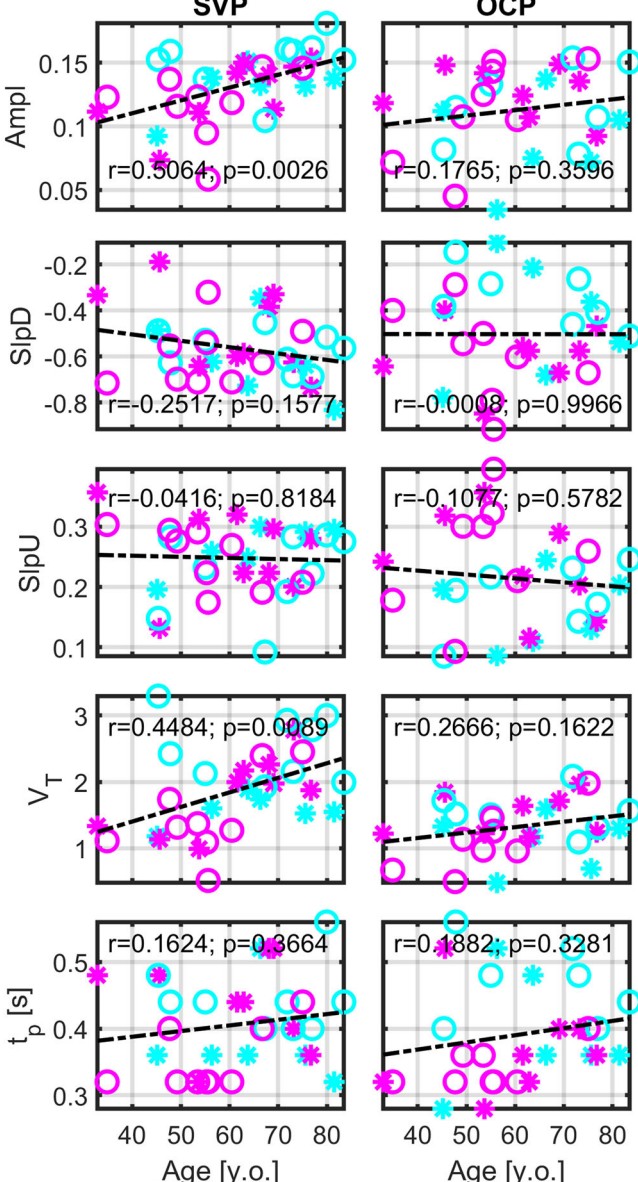

**Fig. 5 Evaluation of linear dependence between age and morphology of retinal pulsatile patterns.** Value r represents a corresponding Pearson correlation coefficient and value p is the *p*-value of the correlation level. SVP spontaneous venous pulsations, OCP optic cup pulsations, Ampl amplitude of the single-pulse in eigenvector space, $V_T$ total relative pulse stroke volume in the eigenvector measures, SlpD slope-down from the eigenvector value at the period beginning to the negative eigenvector peak ≈ peak of the maximal absolute blood volume time-point, SlpU slope-up from the negative eigenvector peak to the period end, $t_p$ time to the negative eigenvector peak.

the Erlangen Glaucoma Registry cohort (www.clinicaltrials.gov, NCT00494923) founded in 1991.

Participants enrolled in the NCT00494923 trial met the criteria of age range 18–65, open chamber angle and corrected visual acuity 0.7 or better when entering the registry. The trial excluded people with systemic disease and potential ocular involvement (e.g., diabetes mellitus), people with myopic or hyperopic refractive error >8.0D, and people with an eye disease (except for glaucoma). From the registry, healthy controls and patients with a history of ocular hypertension (OHT; i.e., intraocular pressure IOP > 21 mmHg, normal visual field and ONH appearance) were included in this study. Pharmacological treatment along with eye surgical intervention history are described in Table 2.

Monocular or binocular retinal video-recordings (RVRs) were acquired for 16 OHT patients (age 58.7 ± 12.9 years old, seven females) and 14 healthy controls

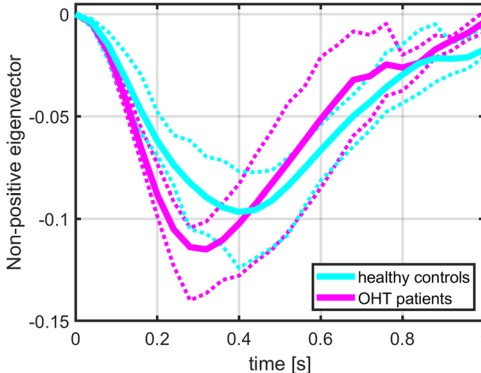

**Fig. 6 Group averaged OCP pulses with 25–75% confidence intervals.** Confidence intervals are visualized as color-matched dashed lines.

(age 66.0 ± 13.2 years old, eight females) between January 2015 and December 2017. In three OHT patients and one healthy control, two RVRs (one monocular and one binocular) were obtained approximately two years apart. In total, 34 RVRs were acquired and analyzed.

Along with RVRs, refractive error, visual acuity, perimetry, IOP, and retinal nerve fiber layer (RNFL) thickness were obtained for each eye with standard clinical devices (white-on-white perimetry with computerized static projection perimeter, Octopus 500, Haag-Streit; Goldmann tonometry; Spectralis OCT, *Heidelberg Engineering*) followed by RVR acquisition. All measurements were acquired within a one-day session.

**Video-ophthalmoscopic data acquisition**. Each participant was examined while comfortably sitting with head rested and positioned on a video-ophthalmoscope chin holder to minimize participant's motion. Each participant was asked to fixate the eyesight at the target presented as a red LED light or cross in the video-ophthalmoscope optical path. The principles of image acquisition were previously described[16]. In short, both available video-ophthalmoscope types (monocular and binocular) acquire images of the reflected light intensity modulated by heart rate induced attenuation changes[16,26]. Because such changes are caused by spatial-temporal retinal blood volume changes, the lowest pixel image intensity corresponds to the highest blood volume (and the highest attenuation) and vice versa.

Left eye monocular RVRs were acquired with the monocular video-ophthalmoscope consisting of the optical lens system (40D ophthalmic lens, two achromatic lenses), one monochrome CCD camera (UI-2210 SE-M-GL, USB interface, *iDS, Germany*), red LED forming the fixation target, and a low power narrow-band LED (wavelength λ = 575 nm) illuminating retina with 30 μW/cm². Acquired 10 s video sequences were saved in non-compressed AVI format with 25 fps (frames-per-second) and matrix size 640 × 480 pixels covering 20° × 15° field of view (i.e., 1 pixel ≈ 9.3 × 9.3 μm²[16,26,51]).

Binocular RVRs were acquired with the binocular video-ophthalmoscope consisting of two optical lens systems, two synchronized CMOS cameras (UI-3060 Rev 2, USB 3.0, *iDS, Germany*), a green OLED display presenting a fixation target, and two narrow-band LED low-power light sources (λ = 575 nm) illuminating retina with 30 μW/cm². Acquired 10 s RVRs were saved in non-compressed AVI format with 25fps and matrix size 1000 × 770 pixels covering 20° × 15° field of view (i.e., 1 pixel ≈ 6.0 × 6.0 μm²)[26,51,52]. Left eye RVRs were only used in further analyses.

**Image analysis**. The whole image analysis workflow is summarized in Fig. 1a. Motion artifacts in RVRs were suppressed with rigid image registration optimized for the RVRs[53]. A representative example of aligned monocular RVR is available at https://youtu.be/-CABIpjWX8Y and binocular RVR at https://youtu.be/4anapI0TZTQ. Next, the optic nerve head (ONH) area was manually segmented from the averaged aligned RVR image and defined as a region of interest (ROI) for further data analysis. Each ONH time frame was spatially smoothed with a 3 × 3 uniform convolution kernel to increase the signal-to-noise ratio (SNR) between local retinal hemodynamics and additive Gaussian noise in RVRs. Acquired relative blood volume changes were high-pass filtered in spectral domain with the cut-off frequency 0.12 Hz in each aligned pixel belonging to the ONH. The filter suppressed DC component and low-frequency drift but preserved pixel-specific pulsatile variance.

Spatial principal component analysis[26] (PCA) was estimated via singular value decomposition (Eq. 1) on each preprocessed ONH RVR.

$$\mathbf{X} = \mathbf{U}\mathbf{\Sigma}\mathbf{V}^{\mathrm{T}} \qquad (1)$$

Formation of PCA input (i.e., ONH RVR matrix **X** of m rows and $n_t$ columns where m is a number of analyzed pixels and $n_t$ is a number of time points) and extraction of PCA outputs from left and right eigenvector matrices **U** and **V** are

**Table 2 Eye drop medication and surgical eye intervention in each participant with ocular hypertension.**

|  | Active substance | Class | Surgical intervention |
|---|---|---|---|
| 1 | Timolol | Non-selective beta-blocker | None |
|  | Sodium hyaluronate | Viscosupplementation agent |  |
| 2 | None | – | None |
| 3 | Latanoprost | Prostaglandin analogue | None |
| 4 | None | – | None |
| 5 | Timolol | Non-selective beta-blocker | None |
|  | Dorzolamide | Carbonic anhydrase inhibitor |  |
| 6 | Timolol | Non-selective beta-blocker | None |
|  | Bimatoprost | Prostaglandin analogue |  |
| 7 | Latanoprost | Prostaglandin analogue | None |
| 8 | Bimatoprost | Prostaglandin analogue | None |
| 9 | Timolol | Non-selective beta-blocker | Selective laser trabeculoplasty |
|  | Bimatoprost | Prostaglandin analogue |  |
| 10 | Pilocarpin | Non-selective muscarinic agonist (M3 selectivity on the iris sphincter) | None |
| 11 | Timolol | Non-selective beta-blocker | Laser (chamber angle) |
|  | Clonidine | Alpha-2 adrenergic agonist (affinitypredilection of 200:1 for α-2 versus α-1 receptors) |  |
| 12 | None | – | None |
| 13 | None | – | Selective laser trabeculoplasty |
| 14 | Latanoprost | Prostaglandin analogue | Selective laser trabeculoplasty |
| 15 | Timolol | Non-selective beta-blocker | None |
|  | Latanoprost | Prostaglandin analogue |  |
| 16 | None | – | Laser (other) |

The table describes the active substances used to treat high intraocular pressure with the classification of the substance.

briefly summarized in Fig. 1b. Left eigenvector matrix $U$ of $m$ rows and $n_t$ columns is a matrix of principal components in descending order of corresponding eigenvalues in diagonal square matrix $\Sigma$ of dimensions $n_t$. One principal component is one column vector of $m$ samples in the matrix $U$. Square right eigenvector matrix $V$ of dimensions $n_t$ consists of column eigen time-courses characteristic for the principal component of a matched column in the matrix $U$. (Illustrative lecture of these singular value decomposition basics is available online[54]).

Z-scored spatial principal components (Fig. 1b) were thresholded to zero in each pixel where $|Z| < 1$[26]. The SVP[26] and OCP[26] spatiotemporal patterns were visually identified as a single component for each pattern from a set of the first 12 principal components. PCA eigenvectors characterizing pulsation time-courses (Fig. 1b) were de-trended, and outlier values were restored utilizing the k-means clustering algorithm, as both implemented and fully described in Labounkova et al.[26]. Control points defining continuous part of an RPP time-course with high SNR (see OCP control points in Fig. 1a) were automatically identified for each SVP or OCP eigenvector[44] (Fig. 1b) characterizing the relative blood volume changes[26], and minor manual edits were done if needed. The automated identification of the control points is detailly described in the "Appendix C" of Labounkova et al.[26].

Each RVR segregated several SVP or OCP single pulse repetitions whose beginning and end were well defined by the control points (Fig. 1a). Averaged single SVP or OCP pulse waveforms were derived for each RVR, and quantitative parameters describing its morphology were evaluated. The evaluated morphology parameters were pulse amplitude (Ampl; Fig. 1a), total relative pulse stroke volume as $V_T = -\int_0^T V(t)dt$ (where $V(t)$ is a volume in non-positive eigenvector values, $T$ is pulse period, and $t$ is time; Fig. 1a), slope-down to peak (SlpD; Fig. 1a), slope-up to baseline (SlpU; Fig. 1a), and time to peak ($t_p$; Fig. 1a). The averaged RVR HR = $60/T[\min^{-1}]$ was estimated from averaged SVP and OCP periods T[s].

The Ampl measurement is proportional to the maximum blood volume in the examined ROI during the cardiac cycle. $V_T$ is proportional to the quantity of total blood volume change in the ROI during the cardiac cycle. SlpD is proportional to the steepness of the blood volume filling in the ROI during the cycle and SlpU to the steepness of blood volume drainage outside the ROI. $t_p$ is proportional to the time of blood volume filling in the ROI during the cycle.

**Statistics and reproducibility**. Delay between overlapping high-quality SVP and OCP portions (Fig. 1a) was evaluated with maximal cross-correlation response function[55]. Pearson correlation coefficients between unaligned or aligned SVP and OCP patterns were quantified (Fig. 1a). The null hypothesis that SVP-OCP delay equals to 0 was tested with one sample $t$-test.

Pearson correlation coefficients and corresponding correlation p-values were evaluated between SVP or OCP morphological features (i.e., Ampl, $V_T$, SlpD, SlpU, and $t_p$; Fig. 1a) and age, HR, IOP, and average RNFL thickness, respectively. Ten correlation effects were investigated for each variable (e.g., age, HR). False discovery rate correction ($p_{FDR} < 0.05$) was applied to the correlation p-values, and Benjamini-Hochberg adjusted p-values ($p_{BH}$) were computed to minimize the risk

of the error type I. Due to the limited sample size, power was estimated for each significant correlation to assess the risk of error type II[56]. Two-sample t-test evaluated between-group differences in RPP morphology measurements, age, HR, IOP, and RNFL. ANCOVA (analysis of covariance) was the second statistical test evaluating between-group differences when HR or age were used as confounding variables. Results of two-sample t-tests and ANCOVA tests were compared, and the effects of HR or age on final between-group results were evaluated (Fig. 1a). Due to the limited sample size, we considered uncorrected $p < 0.05$ significant for t-tests and ANCOVA.

**Reporting summary**. Further information on research design is available in the Nature Research Reporting Summary linked to this article.

## Data availability

De-identified data sets can be made available upon a reasonable email request to *Ivana Labounkova* (ilabounk@umn.edu), *Dr. Folkert Horn* (folkert.horn@augen.imed.uni-erlangen.de), or another responsible personnel from the *Department of Ophthalmology and University Eye Hospital, Friedrich-Alexander University Erlangen-Nürnberg at Erlangen, Erlangen, Germany.*

## Code availability

The MATLAB R2017b programming environment (*MathWorks, Natick, USA*) with academic license and the open-source Retina Imaging Toolbox (https://github.com/umn-milab/retinaimagingtoolbox; GNU GPL version 3 license) were used for all image and statistical analyses and visualizations.

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

## Acknowledgements

The authors thank Dr. Folkert Horn and other staff of the Department of Ophthalmology and University Eye Hospital, Friedrich-Alexander University Erlangen-Nürnberg at Erlangen for their clinical support and information provided to our research team during this study. Also, the authors thank the Center for Behavioral Development and the Center for Magnetic Resonance Research (CMRR) at the University of Minnesota for providing the lab space and computational support. The study was supported by several grand awards: Brno University of Technology grant n. FEKT-S-17-4487, Czech Science Foundation, project no. 21-18578 S, Deutsche Forschungsgemeinschaft (DFG) grant number DFG TO 115/3-1, and "Progressive" grant, stage 3 from the Department of

Pediatrics and the Winston & Maxine Wallin Neuroscience Discovery Fund, University of Minnesota. All funding is highly acknowledged.

## Author contributions

I.L. designed analysis, analyzed data, interpreted results, designed figures, drafted and revised manuscript. R.L. designed analysis, analyzed data, interpreted results, designed figures, drafted and revised manuscript. R.K. designed the study, provided materials, revised and proofed the manuscript. R.P.T. designed the study, acquired and provided data, revised and proofed the manuscript. C.F.B. verified, interpreted, and discussed physiology of results, revised and proofed manuscript. C.M.M. verified study design, discussed potential ophthalmological impact and applications, revised and proofed manuscript. B.R.M. discussed potential cerebrovascular impact and applications, revised and proofed manuscript. I.N. supervised and organized the whole team, provided materials, led the discussion, drafted and revised the manuscript.

## Competing interests

The authors declare no competing interests.
