## [Peer Review File · Communications Biology]

Redactions – unpublished data: Parts of this Peer Review File have been redacted as indicated to maintain the confidentiality of unpublished data.Reviewers' comments:

Reviewer #1 (Remarks to the Author):

The major claim of the MS is that HR and age are confounders that should be considered when considering future studies. The study/paper sections have been composed according to the journal guidelines. While there is no fundamental novelty in the study and the results suffer from a small sample size, the study would be of interest to the research community. Specifically, age is an expected and well-known confounder, and HR is also known to be affected by aging. However, the authors have demonstrated the in-vivo evidence that HR and age modulate the morphology of retinal pulsatile patterns in humans, corroborating Levine and Bebie's theory. A larger cohort, still, is needed to verify these preliminary results. There are still points authors should document better for better interpretability of their study/results:

1. The optic nerve head (ONH) area was manually segmented. However, it is unclear whether 2 or more graders evaluated the performance of the segmentation. How the results may have been affected if intra-rater/inter-rater variability is not acceptable?
2. Lower sample sizes are associated with increased variability and a lower probability of replication, and promising-looking correlations are easily obtained by chance when sample sizes are small. The authors should consider that p values near 0.05 typically only provide weak evidence against the null. Why effect size was not calculated (e.g., Cohen's d)?

Reviewer #2 (Remarks to the Author):

Thank you for asking me to review this very interesting paper. The material is very relevant to further understanding of retinal vascular pulse physiology.

I have several broad questions:

1. I had trouble understanding how the principal component analysis was implemented, mainly around the specific data used. I could not see how the SVP data acquisition was separate to the OCP. I assumed that an operator drew around the central veins (after alignment etc) in the case of SVP and the image intensity data from that specific ROI was used. (Like in reference 21) Similarly I assumed that an operator manually drew around the optic cup and that subset of data was used. Also, could the authors describe what the eigenvalues refer to: are they the coefficients of the mean intensity values received by the camera (from the reflected 575 nm light)?
2. The data for the SVP looked very much like data from reference 21, eg fig 2. Could the authors briefly describe the differences as the former (ref 21) data was transformed intensity data integrated over a manually segmented venous region - and intuitively easier to follow (you can see that I am not really a mathematician).
3. The OCP data, I assume included both arterial and venous components (I had to make this assumption because the methods were not clear) and as such the timing and the amplitude etc calculations probably reflect a combination of both integrated over the cup area. Could the authors explain and clarify this. If my assumptions are correct then this would help to explain why the HR was not so strongly associated with OCP features (due to the arterial dominance).
4. Statistical analysis: It all appeared to be univariate analysis to me. I doubt whether the authors can state the independent effect of age upon amplitudes etc (independent of HR) without doing multivariate analysis. I was not sure from the manuscript whether their ANCOVA would allow this, or was performed in this manner. Personally, I would have considered a linear mixed model to allow the use of right and left eye data (to increase power) and in a multivariate manner. In short, this is a useful addition to the literature if it can be made clearer.

Reviewer #3 (Remarks to the Author):

This is an impressive deep dive into retinal vessel pulsatility. I have only a few questions:

Participants: How was OHT defined? Inclusion/exclusion criteria? What was the medication history of all participants (including the drugs used to treat the OHT)?

RVR: presumably the authors used their own custom-built system rather than any commercial device to record these videos? Were the pupils dilated before recording? As the analysis is not entirely automated, please outline which steps require manual input and what effect this has on the outcome measures?

SVP & OCP detection rate: does this depend on the morphology of the optic disc? (eg cup-disc ratio). Can this technique elucidate the same outcome measurements in small crowded optic discs that have no cup?

P5 last para: this is a really important paragraph yet is poorly expressed and difficult to follow – rewrite?

OHT & HR: can the authors speculate on why OHT patients in their study had a mean HR > 10bpm higher than HCs? Does this tie in with the vascular models of OHT/POAG? Has this been reported by others? Is it 'real' or is it merely an artefact of small sample size?

SVP & HR: is it possible that HR does not per se directly influence SVP morphology, but instead both are modulated by circulating factors such as pO₂, pCO₂, pH, catecholamines etc?

Responses to Reviewers

Title: Heart rate and age modulate retinal pulsatile patterns

Editor summary:

You will see that all require more methods details and also some changes in statistical analysis given the small sample size. Similarly, we require additional discussion to be included that clearly states the limitations of the study given this sample size. In general, the reviewers also all felt that the manuscript needed to be much clearer in general.

We thank the Editor and the reviewers for the reviews of our manuscript. We believe that by addressing the comments, the manuscript is clearer and enriched. The responses to the reviewers' comments on a point-by-point basis are below and all the changes made in the manuscript are highlighted by yellow color. The study sample size is the first deeply discussed limitation of the current study.

Reviewers' comments:

1. **Reviewer #1** (Remarks to the Author):

The major claim of the MS is that HR and age are confounders that should be considered when considering future studies. The study/paper sections have been composed according to the journal guidelines. While there is no fundamental novelty in the study and the results suffer from a small sample size, the study would be of interest to the research community. Specifically, age is an expected and well-known confounder, and HR is also known to be affected by aging. However, the authors have demonstrated the in-vivo evidence that HR and age modulate the morphology of retinal pulsatile patterns in humans, corroborating Levine and Bebie's theory. A larger cohort, still, is needed to verify these preliminary results. There are still points authors should document better for better interpretability of their study/results:

- 1.1. The optic nerve head (ONH) area was manually segmented. However, it is unclear whether 2 or more graders evaluated the performance of the segmentation. How the results may have been affected if intra-rater/inter-rater variability is not acceptable?

We thank the reviewer for the comment. ONH areas were manually segmented by one rater and checked by collaborators. The precision of the region of interest (ROI) segmentation is not a critical factor for spatial principal component analysis (PCA), which accumulates eigenvector, i.e., time-course, as a weighted linear mixture over supra-threshold pixels of the given spatial pattern, i.e., spontaneous venous pulsation (SVP) or optic cup pulsation (OCP).

To convince you scientifically, we have evaluated the impact of ROI precision and size on morphology measurements and showed it in this response. Three concurrent ROIs were defined as modified original ROI, i.e., eroded, dilated, and rectangular ROIs (**REDACTED**). The utilized erosion/dilation step was 15 pixels leading to masks with the spatial agreement of $81.5 \pm 4.4\%$ and $83.8 \pm 3.4\%$ to the original ROI, respectively. These levels of agreement represent a larger simulation error than expected to be caused by the inter-rater reading. The last ROI was a circumscribed rectangle to the dilated ROI with a level of agreement of $68.8 \pm 3.0\%$. Varying ROI did not impact high z-score spots of SVP and OCP patterns (**REDACTED**). Varying ROI might have a higher impact on the OCP morphological measurements than on the SVP morphology (**REDACTED**). However, group analysis over all 34 retinal video-recordings demonstrated a minimal

impact of investigated ROIs on final morphological measurements (**REDACTED**). Averaged absolute values of relative error in morphological measurements were about $2.66 \pm 1.78\%$, while of variation were about $29.98 \pm 8.48\%$. In other words, imperfect ROI segmentation would lead to measurement errors $< 5\%$, while measurements of $\approx 95\%$ normally distributed samples would belong to an interval of mean $\pm 60\%$ of the mean. Although paired t-test identified two significant differences when compared to the original ROI (**REDACTED**), neither result passed the multiple comparison errors (i.e., $p < 0.05 / 3$ tests per measurement $\Rightarrow p < 0.0167$).

We very appreciate this reviewer's comment as it motivates a methodological follow-up study investigating the robustness of SVP/OCP morphological measurements via PCA.

- 1.2. Lower sample sizes are associated with increased variability and a lower probability of replication, and promising-looking correlations are easily obtained by chance when sample sizes are small. The authors should consider that p values near 0.05 typically only provide weak evidence against the null. Why effect size was not calculated (e.g., Cohen's d)?

We agree with the reviewer's comment on the lower sample size and variability. Cohen's d was not considered because it assesses the difference between means of two samples of one random variable. This is not the case here. Our input data into Pearson correlation analysis only utilized one sample of two random variables. That indicates that the correlation would still remain significant even if their means differed.

Yet, we are aware of the sample size issue. Therefore, we adjusted the uncorrected p-values at Benjamini-Hochberg p-values and considered significant only the correlation coefficients with $p_{FDR} < 0.05$ (FDR - false discovery rate correction). The utilized FDR correction should minimize the risk of Type I errors, i.e., false positive observations. We have extended results with estimated power for each significant correlation coefficient to assess the risk of Type II errors.

2. Reviewer #2 (Remarks to the Author):

Thank you for asking me to review this very interesting paper. The material is very relevant to further understanding of retinal vascular pulse physiology. I have several broad questions:

- 2.1. I had trouble understanding how the principal component analysis was implemented, mainly around the specific data used. I could not see how the SVP data acquisition was separate to the OCP. I assumed that an operator drew around the central veins (after alignment etc) in the case of SVP and the image intensity data from that specific ROI was used. (Like in reference 21) Similarly I assumed that an operator manually drew around the optic cup and that subset of data was used. Also, could the authors describe what the eigenvalues refer to: are they the coefficients of the mean intensity values received by the camera (from the reflected 575 nm light)?

We thank the reviewer for pointing this out. Field of view (FOV) appeared as shown in [REDACTED] with acquired retinal hemodynamic changes in each image pixel. FOV was reduced to ROI. As originally written in the manuscript methods and shown in Fig. 1, the ONH ROI was manually segmented after the motion artifact correction from the averaged video image. Data of all pixels in the ROI were input into the spatial PCA. The spatial PCA itself blindly separates as much different statistically orthogonal spatial patterns, e.g., the SVP or the OCP, as the dimensionality of the input dataset is. For example, in a dataset with 256 temporal samples for each pixel, the maximal dimensionality of the dataset is 256 dimensions. There was no additional manual drawing of vessels of interest with integrative intensity extraction. The spatial PCA automatically isolates statistically orthogonal patterns (e.g., SVP and OCP) inside the whole ONH ROI. You can consider the process as automated and blind functional sub-parcellation of the ONH. Most of the details have been described in the former study Labounkova et al. (2021)¹ (ref. 26 in manuscript). For higher quality and easier result reproducibility, we have extended methods in this manuscript with singular value decomposition description and more precise references than in the former study. In addition, we have extended Figure 1 with a block-designed PCA input-output system (see below Figure 1b). New Figure 1b should make the insight into the PCA method applied in the manuscript clearer and more transparent. For full details and in addition to ref. 26 (Labounkova et al. 2021¹), we added a reference to a video describing singular value decomposition.

Unlike ref 21 (Morgan et al. 2014²), eigenvalues and eigenvectors do not refer to mean intensity values of the whole marked vein/region of interest. Eigenvalues describe the level of data variance along the eigenvectors, i.e., strength/significance of the component. While eigenvectors described in our paper represent an eigen time-course corresponding to a certain principal component calculated from the 3D input intensity matrix. We have extended the “Methods: Image analysis chapter” with the explanation.

- 2.2. The data for the SVP looked very much like data from reference 21, eg fig 2. Could the authors briefly describe the differences as the former (ref 21) data was transformed intensity data integrated over a manually segmented venous region - and intuitively easier to follow (you can see that I am not really a mathematician).

We thank you for this comment. PCA automatically ranks the input data (Figure 1b) at the linear mixture of statistically orthogonal principal components weighted via eigenvector matrix. The approach is model-free with no manual intervention in the PCA estimation itself. The spatial and temporal patterns (i.e., SVP / OCP and their time-courses) self-organize themselves from the input

data. No manual drawing of SVP ROI was necessary, as described in reference 21 (Morgan et al. 2014²).

Compared to the classic integration method, the advantages of PCA are discussed in detail in ref. 26 in the manuscript (Labounkova et al. 2021¹). Another common approach investigating SVP is to fit a simplified Fourier series model on dynamic retinal data (Betz-Stablein et al. 2018³ and noted ref. 21 – Morgan et al. 2014²). Here, the utilized PCA might also be considered advantageous as the approach is model-free, and the time-course of each principal component is derived in a fully data-driven manner.

- 2.3. The OCP data, I assume included both arterial and venous components (I had to make this assumption because the methods were not clear) and as such the timing and the amplitude etc calculations probably reflect a combination of both integrated over the cup area. Could the authors explain and clarify this. If my assumptions are correct then this would help to explain why the HR was not so strongly associated with OCP features (due to the arterial dominance).

Thank you for this comment. We agree with the reviewer that the manuscript lacked information on the methodology. Therefore, we added this information on the SVP and OCP in the workflow Figure 1 and the expanded methodology section of the manuscript.

As shown in Figure 1 in this manuscript and Fig. 1&2 in ref. 26 (Labounkova et al. 2021¹), it appears that the detected OCP is primarily in the temporal ONH and large vessels. We suppose that the OCP in the temporal part of ONH mainly originates from the posterior ciliary artery supply, as demonstrated in the previous anatomy literature^{4,5}. But, despite the arterial dominance, the venous component (microvessels and large vessels) cannot be excluded, and we agree with the reviewer that our HR associations for OCP features may be affected.

VO is based on the change in the light absorption varying with the blood presence in the retina and may detect the blood volume change in microvessels⁶. Fluorescein and optical coherence tomography angiography (OCT-A) in vivo studies or a methylmethacrylate corrosion casting study on enucleated eyes demonstrated ONH blood supply originating from posterior ciliary artery.⁷⁻⁹ These results further support our assumption about the etiology of cup pulsations. Similar findings were shown in non-human primate studies showing the pre-retinal-arterial filling of the choroid and prelaminar region of OD but not the retinal vascular bed⁷.

Figure 1. Image and statistical analysis workflow. **a)** The block diagram summarizes the whole image and statistical analysis workflow. *Abbreviations:* ONH – optic nerve head; ROI – region of interest; SVP – spontaneous venous pulsation; OCP – optic cup pulsation; T – period [s]; Ampl – amplitude; SlpD – slope down; SlpU – slope up; t_p – time to peak; V_T – total relative pulse stroke volume; ANCOVA – Analysis of covariance. **b)** The block diagram summarizes the input-output system of the utilized principal component analysis (PCA). The left input side demonstrates array operations forming the PCA input matrix. The right output side demonstrates temporal and spatial extraction of the results. *Abbreviations:* x, y – spatial axes of images; n_t – total number of time points; m – total number of pixels of the ROI in one time point; n_1 – index of the principal component number 1 in the output principal component and eigenvector matrices (first red-highlighted columns), and time point number 1 in the PCA input matrix. Each column in the eigenvector matrix represents an eigenvector time-course of the principal component in the corresponding column of the principal component matrix. In the right bottom corner, the thresholded n_1 principal component of m -samples was reshaped back into the original xy space and overlaid the averaged anatomical background image. All images show recordings of the left eye.

2.4. Statistical analysis: It all appeared to be univariate analysis to me. I doubt whether the authors can state the independent effect of age upon amplitudes etc (independent of HR) without doing multivariate analysis. I was not sure from the manuscript whether their ANCOVA would allow this, or was performed in this manner. Personally, I would have considered a linear mixed model to allow the use of right and left eye data (to increase power) and in a multivariate manner.

We thank you for pointing this out. ANCOVA as a multivariate linear mixture model utilized HR or age as confounding regressors. Both variables have not been used in one ANCOVA regression model since age and HR were significantly correlated, and thus, would not meet a condition of two independent regressors. We preferred to utilize left eye data only, as we only had right eye video-recordings for approximately half of our dataset. The dataset sample size is discussed in the study limitations. The added power analysis implies that the risk of the type II error is relatively low in most of our reported correlation measurements. Therefore, we justifiably report the importance of HR and age as confounding variables in between-group inferences.

To the best of our knowledge, we have not noticed any video-ophthalmological study considering the impact of HR in clinical design or statistical analysis. The presented manuscript represents a pilot message to the community: "Be aware of it." Yet, we agree that a larger (ideally multi-center) dataset is needed for the final validation of the HR and age impact. This fact is also discussed in the current manuscript's study limitations and future directions.

2.5. In short, this is a useful addition to the literature if it can be made clearer.

We have added a block diagram (Fig. 1b) summarizing the input-output system of the principal component analysis in the application on retina hemodynamic recording. In addition, we have extended method text in the paragraph describing PCA utilization. And, we have added eigenvector definition in relation to original image intensity fluctuation. We hope that these changes increase the readability of the manuscript.

3. Reviewer #3 (Remarks to the Author):

This is an impressive deep dive into retinal vessel pulsatility. I have only a few questions:

3.1. Participants: How was OHT defined? Inclusion/exclusion criteria? What was the medication history of all participants (including the drugs used to treat the OHT)?

We thank the reviewer for requesting this information. Participants with ocular hypertension had a history of intraocular pressure exceeding 21 mmHg. We have extended the chapter Methods – Participants with characteristics of the participants, i.e., eligibility criteria, inclusion/exclusion criteria. We have added that the information can also be found at www.clinicaltrials.gov, NCT00494923. In the current study, data of healthy controls and OHT participants were only used.

We have also listed detailed medication for each of 16 OHT participants and the past surgical intervention performed on the examined eye (chamber angle correction with laser, laser operation due to other reasons than IOP treatment and anti-glaucoma during cataract extraction) in the newly added Table 2. We are aware that specific agents in eyedrops can alter the ocular hemodynamics with possible effects on the pulse morphology. Therefore, we have discussed this issue/factor in the study limitations paragraph.

- 3.2. RVR: presumably the authors used their own custom-built system rather than any commercial device to record these videos? Were the pupils dilated before recording? As the analysis is not entirely automated, please outline which steps require manual input and what effect this has on the outcome measures?

As stated in the manuscript, the Device was an in-house assembled optical system connected to CCD/CMOS detector (iDS, Germany). The Device was introduced in 2015¹⁰ and has been used for data acquisition of more than a hundred participants since 2015, including glaucoma patients, with peer-reviewed article outputs^{1,11,12}. The range of commercial devices for longer image data acquisition is limited to ophthalmoscopy-based RVA/DVA analyzer^{13,14} (Imedos, Germany), scanning laser ophthalmoscopy built in Spectralis HRA+OCT¹⁵ (Heidelberg Engineering GmbH, Germany) or recent XyCAM device^{16,17}. Except for commercial devices, several concurrent custom-builds exist, such as Doppler holography^{18,19} or Doppler OCT²⁰. The significant advantage of our in-house design is the low-cost assembly and the low light power source enabling “unlimited” recording time in terms of retina safety. Moreover, the in-house assembly offers flexibility in the recording or data analysis while not being bound by proprietary licenses.

In this study, the pupils were dilated before examination with tropicamide as written in Methods. The device does not necessarily require mydriasis, but the video-sequences are of higher quality and signal-to-noise ratio.

Steps requiring manual input

- i. ROI segmentation. We are aware of fully automated algorithms for ONH segmentation and that manual segmentation is a redundant step in this case. In our experience, the automated ONH segmentation did not work well for presented dynamic data. Therefore, we utilized manual segmentation. Our team is working on the optimization of automated ONH segmentation. Once completed, we plan to substitute manual ONH segmentation with an automated one. But as shown in [**REDACTED**], the precision of ONH segmentation did not impact outcome measures. Therefore, we have added a statement about the minimal impact of ONH segmentation on outcome measures in the Discussion.
- ii. The identification of SVP and OCP components was performed manually. The automated identification of components of interest is not only a struggle of the current study or retinal imaging studies, in general. For example, the neuroimaging community also struggles with the same issue over a decade in functional magnetic resonance imaging^{21,22}. Wrong component selection can impact outcome measures so that values are evaluated for an incorrect pattern. We have added this impact into the Discussion.
- iii. The control points in eigen time-courses were calculated automatically, but the manual intervention was sometimes needed. Here the manual intervention improves the precision of the averaged single pulse estimation, and thus, also the precision of outcome measures. Again, we have added the statement into the Discussion.

- 3.3. SVP & OCP detection rate: does this depend on the morphology of the optic disc? (eg cup-disc ratio). Can this technique elucidate the same outcome measurements in small crowded optic discs that have no cup?

We thank the reviewer for the questions. We are aware that SVP pulsation may depend on the ONH morphology, as previously indicated in Hedges et al. (Hedges et al. 1993²³) and confirmed

in (McHugh et al. 2020¹⁵). Unfortunately, we do not have data to support this theory and respond to this question.

3.4. P5 last para: this is a really important paragraph yet is poorly expressed and difficult to follow – rewrite?

We have rewritten the paragraph to ease its readability.

3.5. OHT & HR: can the authors speculate on why OHT patients in their study had a mean HR > 10bpm higher than HCs? Does this tie in with the vascular models of OHT/POAG? Has this been reported by others? Is it 'real' or is it merely an artefact of small sample size?

We strongly believe that our result of higher HR in the OHT group is rather coincidental. In the previous reports of patients with glaucoma or very sporadically with OHT, the patients mainly had lower or no difference compared to healthy subjects. Lower heart rate in both high-tension and normotensive glaucoma was reported compared to healthy controls (Burgansky-Eliash et al. 2016²⁴, Awe et al. 2020). Lower HR in normotensive and high-tension POAG patients may be due to higher parasympathetic cardiovascular activity than in normal individuals (Awe et al. 2022²⁵, Kurysheva et al. 2018²⁶). Findl et al.²⁷ study did not demonstrate any significant difference between glaucoma (n=90) and control groups (n=61). Comparable heart rate values between normal and glaucoma subjects with various levels of severity and treated ocular hypertension patients were shown in the Wang et al. 2015 study²⁸.

On the other hand, we found in the Framingham study²⁹ that a pulse rate may be a predisposing factor of OHT. The pulse rate was measured in the Heart Study in 1948-1964, while the OHT status was diagnosed within the Eye Study in 1973-1975, i.e., 9-25 years after the variable was measured. The study of 52-85 years old showed a higher pulse rate in the OHT group, but only in 65-74-year-old males (mean 72.0 vs. 77.7 bpm) and females (74.6 vs. 78.0 bpm), respectively (not in 52-64 or 75-85 years old). The authors claimed that the findings in this study might be a mix of real and chance associations²⁹.

It is important to note that our OHT subjects had a history of intraocular hypertension. Yet, their IOP levels at the time of VO recordings were normalized and well-compensated with the topical treatment or post-surgery (see newly added Table 2).

3.6. SVP & HR: is it possible that HR does not per se directly influence SVP morphology, but instead both are modulated by circulating factors such as pO₂, pCO₂, pH, catecholamines etc?

Thank you for bringing this complex topic up. In general, and based on the published papers, we suppose that intraocular pressure, retinal venous pressure, and cerebrospinal fluid pressure are the main factors influencing the SVP generation. Thus, we expect that the pressure gradient between the intraocular and retrobulbar (intracranial) space is the primary source of SVP and the heart rate is the predominant driving power.

Additionally, the listed circulating factors and also plasma electrolytes affect the cardiovascular system, including HR, blood pressure, vascular resistance, heart contractibility, and other systemic functions. Undoubtedly, they contribute to the individual cardiovascular physiological variations in healthy individuals. As the participants enrolled in this study were overall healthy (apart from the patient's group with OHT in their medical history), we assume that the circulating factors were also within physiological ranges.

In detail, the superficial vascular plexus of the retina does not respond to the autonomic nervous system as the retina is missing both sympathetic and parasympathetic innervation^{30–33}. However, some studies admitted a certain degree of autonomic control in humans as autonomic nerves in the choroid may supply vessels at the optic nerve head³⁰. Notably, the retinal vascular bed is characterized by autoregulation systems (metabolic, myogenic) to adjust blood flow parameters during transient changes in O₂ saturation or blood gases in general and other metabolites to fill metabolic demands of photoreceptors and preserve normal physiological function. Autoregulation is the ability to maintain constant blood flow despite perfusion pressure changes³².

Blood Gases and pH. The autoregulation system explains the responses to blood gas concentration: hyperoxia causes the constriction of retinal vessels and decreases in blood velocity and flow, while hypercapnia and hypoxia cause vasodilatation^{30,32,33}. As a result, pO₂ is maintained constant during systemic hyperoxia by altering the blood flow. Therefore, hypoxia, hypercapnia, hyperoxia, or hypocapnia are conditions that can impact retinal hemodynamics. Similarly, blood gas concentration is closely related to the inner retina's interstitial pH. In the case of circulatory factors, the pH of the blood has no or minimal influence on retinal blood flow as the experiment demonstrated that the acidification of the blood by injecting HCl or lactic acid affected neither interstitial pH nor retinal blood flow. The experiment implied that interstitial acidosis and not systemic acidosis might be a step in the process inducing the vasomotor response in hypercapnia and hypoxia.³²

Catecholamines: Generally, the retinal vasculature seems to lack autonomic innervation³¹, although some alpha and beta-adrenergic and cholinergic receptors are present³². Nevertheless, in vivo studies of retinal vessel reactivity after systemically administered α 1-adrenergic agonists showed contradictory results. This is mainly due to the systemic blood pressure (BP) change, which explains why it is hard to determine what causes the ocular perfusion pressure changes that are either driven by the pronounced autoregulatory capability of the retinal vascular bed reacting to the systemic BP or (in lesser extent) may also be influenced by direct pharmacological effects³¹.

In summary, the SVP morphology may be influenced by responses of local autoregulation to systemic changes in circulating factors. Still, this topic is very complex; some physiological questions linked to human autoregulation systems are not straightforward. Since our investigated subjects were examined during standard conditions (sitting, at rest, at comfortable room temperature, the same illumination, quiet environment), significant changes in the circulating factors were not expected.

Due to the complexity of this topic beyond the manuscript scope, contradictory results in vessel reactivity to catecholamines, unresolved questions in human retinal autoregulation, and homogenous environmental conditions in our experiment, we have decided not to discuss this topic in the current manuscript.

References

1. Labouneková I, Labounek R, Nestržil I, Odstrčilík J, Tornow R-P, Kolar R. Blind Source Separation of Retinal Pulsatile Patterns in Optic Nerve Head Video-Recordings. *IEEE Trans Med Imaging*. 2021;40:852–864.
2. Morgan WH, Hazelton ML, Betz-Stablein BD, Yu DY, Lind CRP, Ravichandran V, House PH. Photoplethysmographic measurement of various retinal vascular pulsation parameters

- and measurement of the venous phase delay. *Investig Ophthalmol Vis Sci.* 2014;55:5998–6006.
3. Betz-Stablein B, Hazelton ML, Morgan WH. Modelling retinal pulsatile blood flow from video data. *Stat Methods Med Res.* 2018;27:1575–1584.
 4. Hayreh SS. The blood supply of the optic nerve head and the evaluation of it - myth and reality. *Prog Retin Eye Res.* 2001;20:563–593.
 5. Cioffi GA, Buskirk EM. Microvasculature of the anterior optic nerve. *Surv Ophthalmol.* 1994;38:S107–S117.
 6. Tornow R-P, Odstrčilík J, Kolar R. Time-resolved quantitative inter-eye comparison of cardiac cycle-induced blood volume changes in the human retina. *Biomed Opt Express.* 2018;9:6237–54.
 7. Hayreh SS. Blood supply of the optic nerve head and its role in optic atrophy , glaucoma , and oedema of the optic disc. *Brit J Ophthal.* 1969;53:721–748.
 8. Lee KM, Kim JM, Lee EJ, Kim TW. Anterior Optic Nerve Head Perfusion is Dependent on Adjacent Parapapillary Choroidal perfusion. *Sci Rep.* 2019;9:1–8.
 9. Lovasik J V., Gagnon M, Kergoat H. A novel noninvasive videographic method for quantifying changes in the chromaticity of the optic nerve head with changes in the intraocular pressure, pulsatile choroidal blood flow and visual neural function in humans. *Surv Ophthalmol.* 1994;38:S35–S51.
 10. Tornow RP, Kolář R, Odstrčilík J. Non-mydratic video ophthalmoscope to measure fast temporal changes of the human retina. *Eur Conf Biomed Opt Opt Soc Am.* 2015;9540:954006.
 11. Kolar R, Tornow RP, Odstrčilík J, Liberdova I. Registration of Retinal Sequences from New Video-Ophthalmoscopic Camera. *Biomed Eng Online.* 2016;15:57.
 12. Tornow RP, Kolar R, Odstrčilík J, Labounkova I, Horn F. Imaging Video Plethysmography Shows Reduced Signal Amplitude in Glaucoma Patients in the Area of the Microvascular Tissue of the Optic Nerve Head. *Graefes Arch Clin Exp Ophthalmol.* 2021;259:483–494.
 13. Seifertl BU, Vilser W. Retinal Vessel Analyzer (RVA)-design and function. *Biomed Tech (Berl).* 2002;47 Suppl 1:678–681.
 14. Garhofer G, Bek T, Boehm AG, Gherghel D, Grunwald J, Jeppesen P, Kergoat H, Kotliar K, Lanzl I, Lovasik J V., Nagel E, Vilser W, Orgul S, Schmetterer L. Use of the retinal vessel analyzer in ocular blood flow research. *Acta Ophthalmol.* 2010;88:717–722.
 15. McHugh JA, D'Antona L, Toma AK, Bremner FD. Spontaneous Venous Pulsations Detected With Infrared Videography. *J Neuroophthalmol.* 2020;40:174–177.
 16. Rege A, Cunningham SI, Liu Y, Raje K, Kalarn S, Brooke MJ, Schocket L, Scott S, Shafi A, Toledo L, Saeedi OJ. Noninvasive assessment of retinal blood flow using a novel handheld laser speckle contrast imager. *Transl Vis Sci Technol.* 2018;7.
 17. Debuc DC, Rege A, Smiddy WE. Use of XyCAM RI for Noninvasive Visualization and Analysis of Retinal Blood Flow Dynamics During Clinical Investigations. *Expert Rev Med Devices.* 2021;
 18. Puyo L, Pacques M, Fink M, Sahel J-A, Atlan M. In vivo laser Doppler holography of the

human retina. *Biomed Opt Express*. 2018;9:4113.

19. Puyo L, Paques M, Fink M, Sahel J-A, Atlan M. Waveform analysis of human retinal and choroidal blood flow with laser Doppler holography. *Biomed Opt Express*. 2019;10:4942–4963.
20. Wartak A, Beer F, Desissaire S, Baumann B, Pircher M, Hitzenberger CK. Investigating spontaneous retinal venous pulsation using Doppler optical coherence tomography. *Sci Rep*. 2019;9:1–11.
21. Calhoun VD, Adali T. Multisubject independent component analysis of fMRI: A decade of intrinsic networks, default mode, and neurodiagnostic discovery. *IEEE Rev Biomed Eng*. 2012;5:60–73.
22. Griffanti L, Douaud G, Bijsterbosch J, Evangelisti S, Alfaro-almagro F, Glasser MF, Du EP, Fitzgibbon S, Westphal R, Carone D, Beckmann CF, Smith SM. NeuroImage Hand classification of fMRI ICA noise components. *Neuroimage*. 2017;154:188–205.
23. Hedges TRJ, Baron EM, Hedges TRI, Sinclair SH. The Retinal Venous Pulse: Its Relation to Optic Disc Characteristics and Choroidal Pulse. *Ophthalmology*. 1994;101:542–547.
24. Burgansky-Eliash Z, Bartov E, Barak A, Grinvald A, Gatton D. Blood-Flow Velocity in Glaucoma Patients Measured with the Retinal Function Imager. *Curr Eye Res*. 2016;41:965–970.
25. Awe OO, Ogunlade, Oluwadare Adegbehingbe BO. Assessment of Parasympathetic Cardiovascular Activity in Primary Open Angle Glaucoma. *Int Ophthalmol*. 2022;[In Review:1–16.
26. Kuryshva NI, Shlapak VN, Ryabova TY. Heart rate variability in normal tension glaucoma: A case-control study. *Medicine (Baltimore)*. 2018;97.
27. Findl O, Rainer G, Dallinger S, Dorner G, Polak K, Kiss B, Georgopoulos M, Vass C, Schmetterer L. Assessment of optic disk blood flow in patients with open-angle glaucoma. *Am J Ophthalmol*. 2000;130:589–596.
28. Wang X, Jiang C, Ko T, Kong X, Yu X, Min W, Shi G, Sun X. Correlation between optic disc perfusion and glaucomatous severity in patients with open-angle glaucoma: an optical coherence tomography angiography study. *Graefe's Arch Clin Exp Ophthalmol*. 2015;253:1557–1564.
29. Kahn HA, Leibowitz HM, Ganley JP, Kini MM, Colton T, Nickerson RS, Dawber TR. The Framingham Eye Study: II. Association of Ophthalmic Pathology with Single Variables Previously Measured in Framingham Heart Study. *Am J Epidemiol*. 1977;106:33–41.
30. Forrester J V., Dick AD, McMenamin PG, Roberts F, Pearlman E. The Eye: Chapter 4 - Biochemistry and cell biology. In: Forrester J V., Dick AD, McMenamin PG, Roberts F, Pearlman E, editors. . London: Elsevier; 2021. p. 162-274.e6.
31. Ruan Y, Böhmer T, Jiang S, Gericke A. The Role of Adrenoceptors in the Retina. *Cells*. 2020;9:1–21.
32. Kaiser HJ, Flammer J, Hendrickson P. Ocular blood flow: new insights into the pathogenesis of ocular diseases. Basel: Karger Medical and Scientific Publishers; 1996.
33. Schmetterer L, Kiel JW. Ocular Blood Flow. Springer; 2012.

REVIEWERS' COMMENTS:

Reviewer #1 (Remarks to the Author):

The authors have addressed all comments/suggestions. This is very well done study with relevant results.

Reviewer #3 (Remarks to the Author):

thank you for addressing my comments on your original submission, I have no additional comments to make on this revised manuscript